# Deriving Soil Quality Criteria of Chromium Based on Species Sensitivity Distribution Methodology

**DOI:** 10.3390/toxics9030058

**Published:** 2021-03-16

**Authors:** Yuxia Liu, Qixing Zhou, Yi Wang, Siwen Cheng, Weiduo Hao

**Affiliations:** 1Beijing Key Laboratory of Oil and Gas Pollution Control, China University of Petroleum, Beijing 102249, China; liuyuxia_mm@163.com; 2Key Laboratory of Pollution Processes and Environmental Criteria (Ministry of Education), College of Environmental Science and Engineering, Nankai University, Tianjin 300071, China; 13752443261@126.com; 3School of Chemistry and Environmental Engineering, Wuhan Polytechnic University, Wuhan 430023, China; Aa973831917@126.com; 4Department of Earth and Atmospheric Sciences, University of Alberta, Edmonton, AB T6G 2E3, Canada; whao@ualberta.ca

**Keywords:** ecological risk assessment, species sensitivity distribution (SSD), Cr, toxicity

## Abstract

Chromium (Cr) is one of the most severe heavy metal contaminants in soil, and it seriously threatens ecosystems and human health through the food chain. It is fundamental to collect toxicity data of Cr before developing soil quality criteria/standards in order to efficiently prevent health risks. In this work, the short-term toxic effects of Cr(VI) and Cr(III) on the root growth of eleven terrestrial plants were investigated. The corresponding fifth percentile hazardous concentrations (HC_5_) by the best fitting species sensitivity distribution (SSD) curves based on the tenth percentile effect concentrations (EC_10_) were determined to be 0.60 and 4.51 mg/kg for Cr (VI) and Cr (III), respectively. Compared to the screening level values worldwide, the HC_5_ values in this study were higher for Cr(VI) and lower for Cr(III) to some extent. The results provide useful toxicity data for deriving national or local soil quality criteria for trivalent and hexavalent Cr.

## 1. Introduction

Chromium (Cr) is one of the most common heavy metal contaminants in soil, sediments, and groundwater. It occurs in several oxidation states ranging from Cr^2−^ to Cr^6+^, with trivalent (Cr^3+^) and hexavalent (Cr^6+^) states being the most common and stable in terrestrial environments. Both oxidation states are excessively released into the environment due to a variety of industrial applications and anthropogenic activities, such as mining, metallurgy, tanneries, pigment-producing plants, fossil fuel combustion, and chemical fertilizers [1,2,3,4,5,6,7,8]. Chromium can be either beneficial or toxic to plants, animals and humans, depending on its oxidation state and concentrations. At low concentrations, Cr(III) is considered to be an essential component in the metabolism of plants, e.g., the synthesis of amino and nucleic acids [3,9]. For example, Yu and Gu (2007) indicated that 2–5 mg/L Cr(III) increased the transpiration rate and chlorophyll contents of hybrid willow [10]. However, Cr(III) at increased concentrations may prohibit metabolic processes and metalloenzyme systems in living organisms [10,11,12,13]. For example, do Nascimento et al. (2018) reported that 200 mg/kg of Cr(III) in soil increased the expression of *psb*A and *psb*O genes, which were responsible for electron transfer and oxygen evolution and directly influenced the photosynthesis and growth of young cocoa plants [12]. Compared to Cr(III), Cr(VI) compounds have higher solubility and oxidizing potential, and are known as extremely toxic carcinogens that may cause death to animals and humans [11]. Hexavalent Cr complexes can easily cross cellular membranes by means of sulfate ionic channels, and undergo immediate reduction reactions leading to the formation of reactive intermediates, which are harmful to cell organelles, proteins and nucleic acids [13,14,15].

Heavy metal pollutants, such as Cr, have been paid considerable attention because of their toxicity, persistence and biological accumulation, which can generate a serious threat to ecosystems and human health through the food chain. Soil quality criteria are derived based on an upper-limit concentration that shows no undesirable or harmful effects [16,17,18]. It is generally considered that contamination below the soil quality criteria in soil or sediments leads to tolerable or minimal human health and ecological risks in the long-term [16,18,19]. However, countries in the world generally derived the screening level values without distinguishing the chemical valence and fractionation of contaminants. It is not enough for predicting the toxicity of pollutants such as As, Cr, Se and V since these trace metals have different valences with different Eh and pH, and can exist as either cations or anions in the soil environment [4,20,21]. It is obvious that metals existing as cationic species have a greater propensity to associate with the solid and are less bioavailable, while anionic metals enter into the pore-water easily and are more soluble [22].

The species sensitivity distribution (SSD) method is widely used in ecological risk assessment and the development of soil quality criteria. The SSD analysis is based on the statistically cumulative probability distribution of toxicity data for multiple species [23,24]. One function of SSD analysis is to calculate the concentration of a contaminant at which a specified proportion of species will be affected, referred to as the hazardous concentration (HC) for p% of species (HC_p_) [25]. For example, HC_5_ means a point estimation of the hazardous concentration for 5% of species or the 95% protection level [26,27]. The SSD is dependent upon available datasets and can differ in the type of distribution, taxonomic diversity, and sample size.

Vegetables and wheats are an essential part of a healthy and balanced ecosystem, and their phytotoxicity tests are required for risk assessment and environmental monitoring. Phytotoxicity tests include the effects of a contaminant on seed germination, seedling (shoot and root) development, biomass production or other physiological functions [28,29]. It was previously reported that root biomass was the most sensitive tissue to Cr, and Cr predominantly accumulated in plant root tissue, with very limited translocation to the shoot [21,28]. Wong and Bradshaw (1982) indicated that the root growth of *Lolium Perenne* can be ceased due to metal pollution (e.g., Cu, Ni, Mn, Pb, Cd and Cr) [30]. As a result, root growth may be the most rapid and sensitive response to Cr contamination in soil since the root is the first and most direct plant tissue to contact Cr contamination [30]. The aim of this work was: (1) to enrich the database of Cr toxicity and determine the effect of Cr(VI) and Cr(III) on a variety of widely cultured vegetables and crops; (2) to derive the soil quality criteria value of Cr(VI) and Cr(III) based on the SSD method; and (3) to compare the criteria values with other countries worldwide and provide a scientific base for the development of soil quality standards.

## 2. Materials and Methods

### 2.1. Soils and Chemicals

The soil sample was collected from Jixian, Tianjin city, without prehistory of Cr contamination. This region was originally developed from the parental material basalt (Alfisol). In the laboratory, samples were air-dried, sieved through a 5-mm mesh, homogenized using a scoop, and then stored in a glass bottle until their use. Basic soil properties were characterized using standard methods [31] and are shown in Table 1.

Analytical grade CrCl_3_·6H_2_O and K_2_Cr_2_O_7_ (>99% purity) were used as Cr(III) and Cr(VI) sources.

### 2.2. Short-Term Toxicity Tests

Chromium toxicity tests were performed for 11 different crops, including: pakchoi cabbage (*Brassica rapa chinensis*), tomato (*Solanum lycopersicum*), wheat (*Triticum aestivum* L.), chili (*Capsicum annuum* L.), eggplant (*Solanum melongena* L.), celery (*Apium graveolens* L.), chive (*Allium schoenoprasum* L.), lettuce (*Lactuca sativa* L.), cucumber (*Cucumis sativus* L.), radish (*Raphanus sativus* L.), and spinach (*Spinacia oleracea* L.). Among these plants, pakchoi cabbage, lettuce, spinach and chive are leafy vegetables, tomato, chili and eggplant are solanaceous vegetables, celery is a stem vegetable, and cucumber is a gourd vegetable. All plant seeds were purchased from the Chinese Academy of Agricultural Sciences (located in Tianjin city). Before planting, the plant seeds were pre-treated by sterilizing in 3% H_2_O_2_ for 20 min to prevent fungal contamination, washing with sterilized deionized water several times, and then drying softly with tissue (KIMTECH-Clark).

The initial concentrations of Cr(III) and Cr(VI) for test plants are presented in Table 2. The initial concentrations of Cr(III) and Cr(VI) were in accordance with current situations and levels of soil contamination [5,6], and fell into the range to ensure 65% of germination rate for the test plant according to a pilot experiment. About 50 g of soil was weighed and transferred into a culture dish (diameter: 90 mm). Each treatment was performed in triplicate to minimize experimental errors. Each culture dish was planted with 15 sterilized seeds and cultured in an incubator (LRH-250-Gb, made in Guangdong, China). The plants were illuminated with a light/dark cycle of approximately 15/9 h and the temperature was kept around 25 ± 1 °C. All culture dishes were watered and adjusted to 60% of maximal holding capacity with distilled water. Soil without Cr supply was used as a control treatment. When the length of plant roots in the control reached 20 mm and germination rate >65% for each species, the Cr exposure experiment was finished. For each seed, the elongation of both root and seedling exceeding 3 mm was considered to be a successful germination. Generally, the plants need around 2 weeks to meet the standard of successful germination before harvesting. The plants were then removed and rinsed with distilled water. The root elongation of all treatments was measured and calculated. Root architectural traits were analyzed using the software WinRHIZO (WinRHIZO PRP2012, Regents Instruments Inc., Quebec, QC, Canada).

### 2.3. Species Sensitivity Distribution (SSD)

Species sensitivity distribution (SSD) curves were constructed based on short-term toxicity data for 11 terrestrial species. The resulting data were ranked, and the rank percentiles were determined for each data point. Data points with the same concentration were recommended to be assigned separate, sequential ranks, rather than calculating tied ranks [32]. Rank percentiles were calculated using the following equation:(1)$$j=\frac{i}{\left(n+1\right)}\times 100$$
where *j* = rank percentile; *i* = rank of the data point in the data set; *n* = total number of data points in the data set.

Toxicity data such as EC_10_, NOEC (no observed effect concentration), LOEC (lowest observed effect concentration) or MATC (maximum acceptable toxic concentration) were required and fitted to SSD curves. In this study, we determined the SSD curves by both the Slogistic and the Exponential model using software of Origin8.6. The distribution model was fitted to toxicity data points and evaluated with the adjusted coefficient of determination R^2^; the higher the R^2^, the better the goodness of fit.

### 2.4. Statistic Analysis and Data Integration

All statistical analyses were performed by SPSS20.0. Graphs were prepared using Origin8.6. The treatment means were compared using Duncan’s multiple range test (DMRT) and taking *p* < 0.05 as significant. EC values and their confidence intervals were calculated by a log concentration-logit effect regression model, as described previously [33].

## 3. Results

### 3.1. Species Sensitivity to Short-Term Toxicity of Cr

#### 3.1.1. Short-Term Toxicity Tests of Cr(VI)

Short-term toxicity tests of Cr(VI) to eleven terrestrial plants indicated by root growth inhibition are shown in Figure 1 and Table 3. Different plants displayed variable toxic responses to Cr(VI). The leafy vegetables, such as pakchoi cabbage, lettuce and chive, were the most sensitive species to Cr(VI), and the root inhibition rates were positively correlated with the concentration of Cr(VI). Specifically, the root inhibition rate of pakchoi cabbage reached as high as 30% under 2 mg/kg of Cr(VI) treatment, and the root inhibition rate exceeded 50% under 6 mg/kg of Cr(VI) treatment. The EC_10_ and EC_20_ for pakchoi cabbage were lower than 0.65 mg/kg and 1.13 mg/kg, respectively. For spinach, 10 mg/kg of Cr(VI) led to around 60% root inhibition rate, then the inhibition rate fluctuated when the concentration of Cr(VI) increased further. Wheat had relatively lower sensitivity to Cr(VI) compared to leafy vegetables, and the EC_10_ and EC_20_ for wheat were 1.50 mg/kg and 2.25 mg/kg, respectively. By contrast, the solanaceous vegetables, such as chili, tomato and eggplant, showed less sensitivity to Cr(VI), and the root inhibition rate was linearly correlated to Cr(VI) concentration. For example, the Cr(VI) concentration which resulted in a 30% root inhibition rate was lower than 5 mg/kg for all three solanaceous vegetables, and the Cr(VI) concentration which led to a 50% root inhibition rate was around 10 mg/kg. The EC_10_ and EC_20_ for solanaceous vegetables ranged around 1.55–2.78 mg/kg and 2.61–3.71 mg/kg, respectively. Stem vegetables (celery) and gourd vegetables (cucumber) showed the least sensitivity to Cr(VI). The root inhibition rate for celery was not obvious at Cr(VI) concentrations lower than 10 mg/kg. Intrudingly, 10 mg/kg of Cr(VI) supply seemed to enhance the growth of cucumber root. The EC_10_ and EC_20_ for cucumber even exceeded 12.83 mg/kg and 16.62 mg/kg, respectively. The effect of Cr(VI) on the root growth of the above plants generally followed the order of pakchoi cabbage > lettuce > chive > wheat > tomato > radish > chili > eggplant > spinach > celery > cucumber. Leafy vegetables (e.g., pakchoi cabbage, lettuce and chive) might be the primary affected plants under Cr(VI) contamination, while the root growth of stem vegetables (celery) and gourd vegetables (cucumber) might be insensitive to Cr(VI) contamination.

#### 3.1.2. Short-Term Toxicity Tests of Cr(III)

As indicated by our experimental results, the root inhibition rate was positively correlated to Cr(III) concentration, as shown in Figure 2 and Table 4. Pakchoi cabbage and chive showed the most significant responses to Cr(III) toxins, and 50 mg/kg of Cr(III) led to more than 20% root inhibition. A total of 300 mg/kg of Cr(III) led to more than 50% root inhibition for chive and 35% root inhibition for pakchoi cabbage. The EC_10_ and EC_20_ for pakchoi cabbage and chive ranged around 5.62–11.84 mg/kg and 24.21–36.56 mg/kg, respectively. By contrast, solanaceous vegetables were less sensitive to Cr(III) compared to other studied species. For example, eggplant was only sensitive to Cr(III) at concentrations lower than 100 mg/kg, and the root growth was less affected above 100 mg/kg of Cr(III). The EC_10_ and EC_20_ for eggplant were 16.78 mg/kg and 24.71 mg/kg, respectively. The root inhibition rate of tomato was linearly correlated to Cr(III) concentration, with 200 mg/kg of Cr(III) leading to 30% root inhibition and 500 mg/kg of Cr(III) leading to 50% root inhibition. Among the vegetables studied, the root growth of the gourd vegetable (cucumber) and the root vegetable (radish) showed the least sensitivity to Cr(III). For example, 500 mg/kg of Cr(III) supply led to less than 50% root growing inhibition for cucumber, and 500 mg/kg of Cr(III) supply only led to less than 30% root inhibition for radish. The EC_10_ and EC_20_ for radish were 189.10 mg/kg and 328.60 mg/kg, respectively. It is interesting that Cr(III) posed an insignificant impact on the growth of leafy vegetables such as spinach, with 500 mg/kg of Cr(III) supply leading to less than 30% root inhibition, and the EC_10_ and EC_20_ were as high as 113.10 mg/kg and 187.20 mg/kg, respectively. Wheat was also insensitive to Cr(III), with 1000 mg/kg of Cr(III) supply leading to less than 50% root inhibition. The effect of Cr(VI) on the root growth of the above plants generally followed the order of pakchoi cabbage > chive > eggplant > celery > tomato > lettuce > chili > cucumber > wheat > spinach> radish. The root growth of leafy vegetables (e.g., pakchoi cabbage and chive) was easily affected by Cr(III) contamination, while root vegetables (e.g., radish) endured high concentrations of Cr(III) contamination.

### 3.2. Ecological Risk Assessment Based on Species Sensitivity Distributions (SSD)

The details of formulas and parameters for SSD curves based on the Slogistic and the Exponential model are shown in Table 5. The fitted SSD curves for Cr(VI) and Cr(III) are shown in Figure 3 and Figure 4, respectively. For Cr(VI) and Cr(III), two types of SSD curves were fitted: SSD-EC_10_ and SSD-EC_20_. From each SSD-EC_x_ curve, HC_5_ and HC_10_ were numerically derived. The HC_5_ values based on the EC_10_ of the eleven plants were 0.60 mg/kg and 4.51 mg/kg for Cr(VI) and Cr(III), respectively. Furthermore, the HC_10_ values based on the eleven plants were 0.83 mg/kg and 16.98 mg/kg for Cr(VI) and Cr(III), respectively. The HC values for Cr(VI) were far less than Cr(III), indicating the higher toxicity of Cr(VI) to tested plants than Cr(III).

## 4. Discussion

Irrespective of Cr valence, the root growth of leafy vegetables, such as pakchoi cabbage and chive, was easily affected by Cr contamination. By contrast, the root growth of stem vegetables (e.g., celery) and root vegetables (e.g., radish) was less affected by Cr contamination. Cr contaminants generally damaged the root architecture of plants by limiting water and nutrient uptake efficiency [34], and induced a negative impact on the functioning of leaf photosynthetic machinery by reducing Fe and Mg uptake, which are essential for chlorophyll biosynthesis [35,36]. Photosynthesis was the physiological foundation for crop growth and was considered to be one of the most stress-sensitive processes [37]. Furthermore, Cr was found to be stable in roots and poorly translocated from roots to upper tissues. It is reported that the maximum amount of Cr is accumulated in the roots, followed by leaves and then fruits [38]. For radish, the root is the edible part and might accumulate a large amount of Cr by growing under highly Cr-contaminated soil, which will result in a health risk though the food chain.

With respect to the Cr valence, the plant root growth was preferably impacted by Cr(VI) than Cr(III), since Cr(VI) produced lower EC_10_ and EC_20_ than Cr(III) did (Table 3 and Table 4). Accordingly, the HC_5_ value derived from EC_10_ for Cr(VI) was 0.60 mg/kg, which was lower than that for Cr(III) (4.51 mg/kg), demonstrating that hexavalent Cr compounds were more hazardous than trivalent Cr compounds. It was reported that the uptake mechanisms of Cr(III) and Cr(VI) were quite different, Cr(VI) was immediately converted to Cr(III) in roots by Fe(III)-reductase enzymes [11,39,40], and plant tissues endured higher Cr(III) than Cr(VI) [13,40]. Thus, the total Cr contents in soil do not necessarily reflect the biogeochemical behavior of Cr since the biogeochemical behavior of Cr differs with chemical speciation [21].

Cr speciation and its bio-organism transfer were governed by soil physico-chemical properties and microbial activity. From the southeast coast to northwest inland in China, the major soil types cover red soil, brown soil and cinnamon soil, which are quite different in terms of pH, redox conditions, cation exchange capacity, biological and microbial conditions and co-existing competing cations. These soil types will differ significantly in Cr dynamic reactions and behaviors, including hydrolysis, oxidation, reduction and precipitation [20,21], which lead to various Cr speciation and distribution. Shahid et al. (2017) indicated that toxic Cr(VI) might transform to less toxic Cr(III) in the presence of dissolved organic carbon originating from the solubilization of soil organic matter [21]. As a result, organic carbon can reduce Cr bioavailability and detoxify its negative effect on plant growth [40]. It is reported that the soil microbial community played a key role in governing Cr speciation [41], and Cr(VI) was able to interact with microorganisms via enzymatic biosorption, reduction and bioaccumulation [42]. The rate and extent of microbial-mediated Cr(VI) reduction greatly varied with bacterial strain, Cr concentration, as well as soil physio-chemical properties and co-existing contaminants [21,43]. As a result, soil screening values should be directly related to soil properties and conditions.

However, the Soil Environmental Quality Standards of China set the screening level values of Cr as 300 mg/kg for paddy fields and 200 mg/kg for dry land in the pH range of 6.5–7.5. These values are comparable to the values for the total Cr of Australia, the Netherlands and Thailand. A total of 200–300 mg/kg of Cr(III) in soil might not poison terrestrial plants, but 200–300 mg/kg of Cr(VI) exposure will pose a harmful impact on bio-organisms. Therefore, it is necessary to derive and formulate screening level values for different Cr oxidation states. Table 6 lists the soil quality standard values for Cr from different countries, and only a few countries (e.g., the United States, Japan, and Canada) have separated standard values for Cr(VI) and total Cr [22,32,44]. Swedish, German, and Dutch soil quality standards are stringent and trying to minimize the impact on humans and surrounding ecosystems [45,46,47]. Therefore, there is no urgent need to revise existing limit values for these countries. The HC_5_ values of Cr(VI) based on the SSD-EC_10_ and HC_10_ of Cr(VI) based on SSD-EC_10_ in our experiments were higher than the screening level values in Japan, Canada and the United States [22,32,44]. However, the HC_5_ values of Cr(III) based on the SSD-EC_10_ and HC_10_ of Cr(III) based on SSD-EC_10_ were far less than the standard values of Cr in the other countries [45,46,47,48,49]. This may because the EC_10_ was close to acute NOECs or the tested species may be too sensitive to Cr(III). Moreover, the difference in the HC_5_ values between China and other countries might be attributed to the distribution of different resident species [50,51]. The unique taxonomic composition and complexity of ecosystems could lead to the over-protection or under-protection of terrestrial biorganisms therein. This suggests that the development of local soil quality criteria characterizing the sensitive species and potential risk acceptors in China is necessary.

Soil heavy metal pollution has become a severe global problem. In light of a nationwide soil quality and pollution survey launched between 2006 and 2010 in China, overall 16.1% of soil was polluted, of which 82.2% was inorganic pollution, and Cr exceeding rates were 1.1% [52]. In order to enact scientific and practical soil quality standards, comprehensive soil surveys and advanced monitoring techniques are required, and it is essential to acquire the background concentration of elements, environmental quality, nutrient status, and physical and chemical properties of soils in the future. It is also highly practical to monitor the magnitude of the risks involved with the consumption of vegetables cultivated in contaminated soils. In order to have a better protection of the terrestrial ecosystem, it is essential to take the local geological and natural conditions of different land used into consideration. To meet the goals of protecting and conserving soil for sustainable use, a systematic soil quality standard, including national, regional and industrial soil quality, is advised to be developed. Studies on the regionalization of terrestrial ecological functions and potential risk acceptors and the collection of toxicity data will be ongoing.

## 5. Conclusions

The root inhibition rates of tested plants were positively correlated with either Cr(VI) or Cr(III) concentrations. Generally, leafy vegetables were the most sensitive plants to Cr contamination in soils, followed by solanaceous and stem vegetables. Gourd vegetables were the least sensitive plants. The HC_5_ values based on EC_10_ were 0.60 mg/kg for Cr(VI) and 4.51 mg/kg for Cr(III), and the HC_10_ values based on EC_10_ were 0.83 mg/kg for Cr(VI) and 16.98 mg/kg for Cr(III). Compared to the screening level values worldwide, the HC_5_ and HC_10_ values were higher for Cr(VI) and lower for Cr(III) to some extent. It is recommended that a scientific and practical soil quality standard should be developed to characterize the sensitive species and potential risk acceptors, considering chemical valence and species of target contaminants as well.

## Figures and Tables

**Figure 1 toxics-09-00058-f001:**
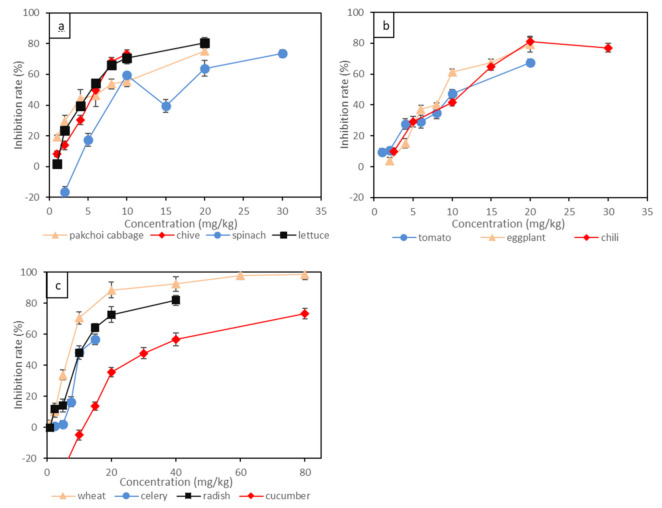
The inhibition rate of Cr(VI) on root elongation of 11 plants: (**a**) leafy vegetables; (**b**) solanaceous vegetables; (**c**) wheat, gourd vegetable and stem vegetable. Bars represent standard error (*n* = 3).

**Figure 2 toxics-09-00058-f002:**
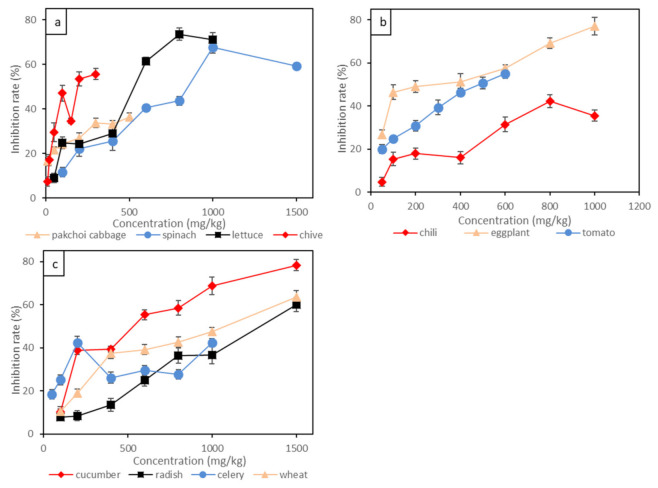
The inhibition rate of Cr(III) on root elongation of 11 plants: (**a**) leafy vegetables; (**b**) solanaceous vegetables; (**c**) wheat, gourd vegetable and stem vegetable. Bars represent standard error (*n* = 3).

**Figure 3 toxics-09-00058-f003:**
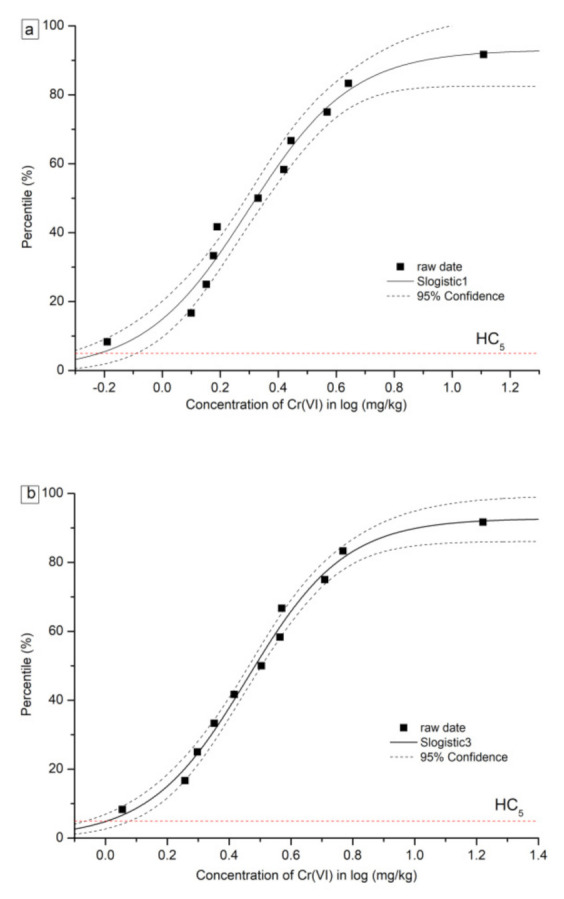
Curves of species sensitivity distribution for Cr (VI) based on (**a**) EC_10_, (**b**) EC_20_.

**Figure 4 toxics-09-00058-f004:**
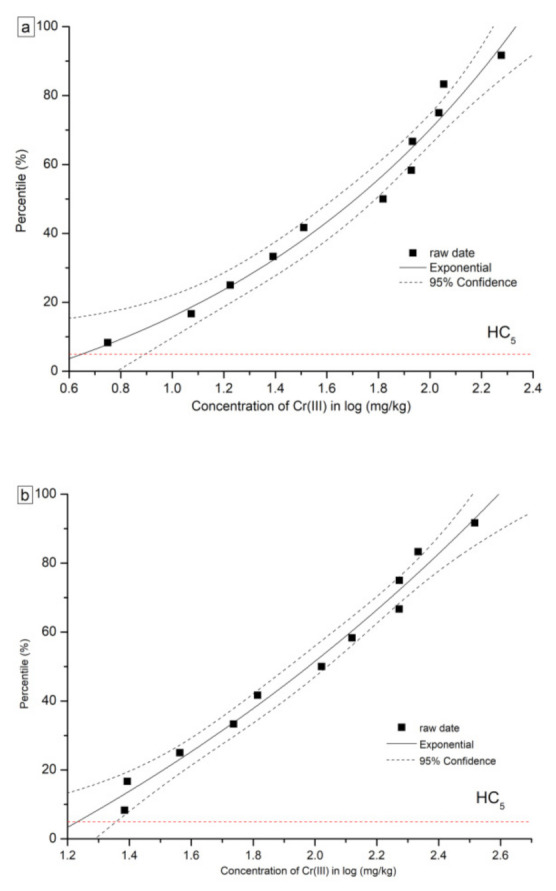
Curves of species sensitivity distribution for Cr (III) based on (**a**) EC_10_, (**b**) EC_20_.

**Table 1 toxics-09-00058-t001:** Basic properties of the tested soil.

pH	Organic Matter (g/kg)	Cation Exchange Capacity (cmol/kg)	Total Nitrogen (g/kg)	Total Phosphorus (%)
7.1	16.77	29.5	1.48	1.48

**Table 2 toxics-09-00058-t002:** Initial concentration of Cr(III) and Cr(VI) for test plants.

Species	Cr(III) mg/kg	Cr(VI) mg/kg
*Brassica rapa chinensis*	0, 10, 50, 100, 200, 300, 400, 500	0, 1, 2, 4, 6, 8, 10, 20
*Lactuca sativa* L.	0, 50, 100, 200, 400, 600, 800, 1000	0, 1, 2, 4, 6, 8, 10, 20
*Allium schoenoprasum* L.	0, 10, 20, 50, 100, 150, 200, 300	0, 1, 2, 4, 6, 8, 10, 20
*Triticum aestivum* L.	0, 100, 200, 400, 600, 800, 1000, 1500	0, 1, 2.5, 5, 10, 20, 40, 60, 80
*Solanum lycopersicum*	0, 50, 100, 200, 300, 400, 500, 600	0, 1, 2, 4, 6, 8, 10, 20
*Raphanus sativus* L.	0, 100, 200, 400, 600, 800, 1000, 1500	0, 1, 2.5, 5, 10, 15, 20, 40
*Capsicum annuum* L.	0, 50, 100, 200, 400, 600, 800, 1000	0, 2.5, 5, 10, 15, 20, 30, 40
*Solanum melongena* L.	0, 50, 100, 200, 400, 600, 800, 1000	0, 2, 4, 6, 8, 10, 15, 20
*Spinacia oleracea* L.	0, 100, 200, 400, 600, 800, 1000, 1500	0, 1, 2, 5, 10, 15, 20, 30
*Apium graveolens* L.	0, 50, 100, 200, 400, 600, 800, 1000	0, 2.5, 5, 7.5, 10, 15, 30
*Cucumis sativus*	0, 100, 200, 400, 600, 800, 1000, 1500	0, 5, 10, 15, 20, 30, 40, 80

**Table 3 toxics-09-00058-t003:** Inhibition effects of Cr(VI) on root elongation of 11 terrestrial plants.

Species	Rank	EC_10_ mg/kg	Rank	EC_20_ mg/kg
*Brassica rapa chinensis*	1	0.65 (0.55~0.76)	1	1.13 (0.97~1.33)
*Lactuca sativa* L.	2	1.26 (1.10~1.44)	2	1.81 (1.58~2.07)
*Allium schoenoprasum* L.	3	1.42 (1.01~2.00)	3	1.98 (1.41~2.80)
*Triticum aestivum* L.	4	1.50 (1.18~1.91)	4	2.25 (1.77~2.86)
*Solanum lycopersicum*	5	1.55 (1.10~2.18)	5	2.61 (1.85~3.67)
*Raphanus sativus* L.	6	2.14 (1.62~2.83)	6	3.19 (2.41~4.22)
*Capsicum annuum* L.	7	2.63 (1.99~3.46)	7	3.67 (2.78~4.83)
*Solanum melongena* L.	8	2.78 (2.26~3.42)	8	3.71 (3.02~4.57)
*Spinacia oleracea* L.	9	3.71 (2.83~4.86)	9	5.13 (3.91~6.71)
*Apium graveolens* L.	10	4.39 (2.30~8.38)	10	5.87 (3.07~11.22)
*Cucumis sativus*	11	12.83 (10.88~15.13)	11	16.62 (14.09~19.61)

Note: Values are given as means with 95% confidence limits.

**Table 4 toxics-09-00058-t004:** Inhibition effects of Cr(III) on root elongation of 11 terrestrial plants.

Species	Rank	EC_10_ mg/kg	Rank	EC_20_ mg/kg
*Brassica rapa chinensis*	1	5.62 (4.62~6.98)	3	36.56 (29.12~46.63)
*Allium schoenoprasum* L.	2	11.84 (8.53~16.45)	1	24.21 (17.44~33.62)
*Solanum melongena* L.	3	16.78 (13.05~22.21)	2	24.71 (16.97~35.98)
*Apium graveolens* L.	4	24.59 (21.02~29.54)	4	54.52 (39.31~75.63)
*Solanum lycopersicum*	5	32.35 (25.00~41.86)	5	65.16 (50.36~84.32)
*Lactuca sativa* L.	6	65.72 (41.86~103.20)	6	105.10 (66.96~165.10)
*Capsicum annuum* L.	7	84.68 (51.90~138.20)	10	215.60 (132.10~351.70)
*Cucumis sativus*	8	85.58 (67.52~108.50)	7	131.70 (103.90~166.90)
*Triticum aestivum* L.	9	108.30 (87.17~134.70)	9	187.40 (150.80~232.90)
*Spinacia oleracea* L.	10	113.10 (74.35~171.90)	8	187.20 (123.00~284.70)
*Raphanus sativus* L.	11	189.10 (115.90~308.30)	11	328.60 (201.50~535.80)

Note: Values are given as means with 95% confidence limits.

**Table 5 toxics-09-00058-t005:** The models and parameters of curve fitting and the values of HC_5_ and HC_10_ for chromium.

Contaminants	Cr(VI)		Cr(III)	
	EC_10_	EC_20_	EC_10_	EC_20_
Model	Slogistic1	Slogistic3	Exponential	Exponential
Formula	y = a/(1 + exp (−k × (x − x_c_)))	y = a/(1 + b × exp (−k × x))	y = y_0_ + A × exp (R_0_ × x)	y = y_0_ + A × exp (R_0_ × x)
Parameter	a = 93.04	a = 92.75	y_0_ = −29.50	y_0_ = −110.64
	x_c_ = 0.30	b = 18.30	A = 20.69	A = 67.19
	k = 5.59	k = 6.33	R_0_ = 0.79	R_0_ = 0.44
R^2^	0.97	0.99	0.97	0.98
HC_5_	0.60	1.01	4.51	6.65
HC_10_	0.83	1.34	16.98	21.33

Note: y is the cumulative probability of species, x is the mean of the log_10_ EC_10_ or log_10_ EC_20_.

**Table 6 toxics-09-00058-t006:** Risk-based screening levels of Cr in some countries (mg/kg).

Country	Cr(VI)	Total Cr	Reference
Australia		100	NEPC, 1999
Canada	0.4		CCME, 2006
Germany		30 (sand), 60 (loam), 100 (clay) ^1^	FSPR, 1999
Sweden		80	Naturvårdsverket, 2009
Netherlands		100	RIVM, 1997
United States	0.29		U.S. EPA, 2010
Japan	0.05 ^2^		MEGJ
Thailand		300	PCD, 2004
Belgium	2.5	34	Carlon, 2007

Note: ^1^ Germany differentiates metal limits according to soil properties; ^2^ The unit is mg/L.

## Data Availability

Data is contained within the article.

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
