# Peer review of "Deriving Soil Quality Criteria of Chromium Based on Species Sensitivity Distribution Methodology"

_toxics, 2021, doi:10.3390/toxics9030058_

Round 1

Reviewer 1 Report

This work elegantly describe the modeled toxicity of Cr (III) and Cr(VI) in different plants, thus analyzing soil quality criteria value based on SSD method. I found this article linear, well-structured, well-written and explained. Methods and Results are well-described. Authors only need to check a few things about the text. 

Line 27: trivalent (Cr III) and hexavalent states (Cr VI)

LIne 31: replace "Cr" with "Chromium"

Line 87: Replace "After air-dried, sieved through a 5-mm mesh, and homogenized thoroughly, the 87 soil was stored in glass bottle for use" with "In the laboratory, samples were air-dried, sieved through 5-mm mesh, homogenized using ...., and then stored in glass bottle until their use".

Line 102: "water" means sterilized deionized water?

LIne 103: the "tissue" was sterilized?

LIne 235: replace "Cr" with "Chromium"

LIne 238: replace "biosyhthesis" with "biosynthesis"

Author Response

Comments and Suggestions for Authors

This work elegantly describe the modeled toxicity of Cr (III) and Cr(VI) in different plants, thus analyzing soil quality criteria value based on SSD method. I found this article linear, well-structured, well-written and explained. Methods and Results are well-described. Authors only need to check a few things about the text. 

YX – We thank the reviewer for his/her positive and helpful comments. We now have answered the reviewer’s comments point by point and revised the manuscript accordingly.

Line 27: trivalent (Cr III) and hexavalent states (Cr VI)

YX – The change has been made accordingly.

LIne 31: replace "Cr" with "Chromium"

YX – The change has been made accordingly.

Line 87: Replace "After air-dried, sieved through a 5-mm mesh, and homogenized thoroughly, the 87 soil was stored in glass bottle for use" with "In the laboratory, samples were air-dried, sieved through 5-mm mesh, homogenized using ...., and then stored in glass bottle until their use".

YX – The change has been made accordingly.

Line 102: "water" means sterilized deionized water?

YX – Thank you for your helpful suggestion, the change has been made accordingly.

Line 103: the "tissue" was sterilized?

YX – The change has been made accordingly.

Line 235: replace "Cr" with "Chromium"

YX – The change has been made accordingly.

Line 238: replace "biosyhthesis" with "biosynthesis"

YX – The change has been made accordingly.

Reviewer 2 Report

My suggestion to authors is to check the significant figures in Tables and the text. Is it meaningful to present so many decimal figures, especially in the toxicity data;

Author Response

Comments and Suggestions for Authors

My suggestion to authors is to check the significant figures in Tables and the text. Is it meaningful to present so many decimal figures, especially in the toxicity data.

YX – We thank the reviewer for his/her positive and helpful comments. We now have checked and changed the decimal of figures and tables accordingly. See changes in revised manuscript with red mark.